# Gene Therapy for Systemic or Organ Specific Delivery of Manganese Superoxide Dismutase

**DOI:** 10.3390/antiox10071057

**Published:** 2021-06-30

**Authors:** Joel S. Greenberger, Amitava Mukherjee, Michael W. Epperly

**Affiliations:** Department of Radiation Oncology, UPMC Hillman Cancer Center, School of Medicine, University of Pittsburgh, Pittsburgh, PA 15232, USA; Amm223@pitt.edu (A.M.); epperlymw@upmc.edu (M.W.E.)

**Keywords:** radioprotection, radiomitigation, plasmid/liposome complex, mitochondrial targeting, MnSOD mimetics

## Abstract

Manganese superoxide dismutase (MnSOD) is a dominant component of the antioxidant defense system in mammalian cells. Since ionizing irradiation induces profound oxidative stress, it was logical to test the effect of overexpression of MnSOD on radioresistance. This task was accomplished by introduction of a transgene for MnSOD into cells in vitro and into organs in vivo, and both paradigms showed clear radioresistance following overexpression. During the course of development and clinical application of using MnSOD as a radioprotector, several prominent observations were made by Larry Oberley, Joel Greenberger, and Michael Epperly which include (1) mitochondrial localization of either manganese superoxide dismutase or copper/zinc SOD was required to provide optimal radiation protection; (2) the time required for optimal expression was 12–18 h, and while acceptable for radiation protection, the time delay was impractical for radiation mitigation; (3) significant increases in intracellular elevation of MnSOD activity were required for effective radioprotection. Lessons learned during the development of MnSOD gene therapy have provided a strategy for delivery of small molecule SOD mimics, which are faster acting and have shown the potential for both radiation protection and mitigation. The purpose of this review is to summarize the current status of using MnSOD-PL and SOD mimetics as radioprotectors and radiomitigators.

## 1. Introduction

Manganese superoxide dismutase (MnSOD2 or SOD2) is a mitochondrial targeted antioxidant whose main responsibility is the dismutation of superoxides produced in the mitochondria during respiration [1,2,3]. MnSOD also has the ability to neutralize reactive oxygen species as well. Irradiation of the cell results in increased reactive oxygen species (ROS) which damages the cells as well as causing increased DNA strand breaks [2,3]. The cell will require increased energy to repair the irradiation damage. Increased antioxidant activity provided by the overexpression of MnSOD in the mitochondria will aid in the preservation of the mitochondria to allow for the increased need of energy in the cell for repair of irradiation damage [1,2,3].

The logic of developing Manganese Superoxide Dismutase (MnSOD) gene therapy arose from the challenge to provide better normal tissue protection during clinical radiotherapy [1,2,3,4]. The basic principles of effective radiotherapy rely upon the concept of the therapeutic ratio: the greater relative level of tumor cytoreduction compared to normal tissue damage [5,6,7]. The principles of clinical radiotherapy also rely upon attention to detail regarding the physical parameters associated with an effective therapeutic ratio, including (1) minimizing the volume of normal tissue irradiated; (2) not exceeding the necessary therapeutic dose delivered; (3) the fraction size (dose given per day over a multiday/multiweek course of radiotherapy); (4) the overall duration (time course in days to weeks) of each particular treatment program. Another consideration is the history of prior irradiation to a particular treatment volume (even if years ago) to estimate the degree of normal tissue toxicity, since irradiation doses can be cumulative. The particular normal tissues and organs in the treated volume are also quite relevant and require adjustment of the therapeutic ratio.

Clinical radiation oncology has evolved, as a medical specialty with its primary goal to provide optimization of the therapeutic ratio to prevent acute and late toxicity [5,6,7]. The ability to shape a radiation field around a cancer target volume, while protecting normal tissues, has been greatly facilitated by improvement in the physical parameters of therapy machines. The use of multileaf collimators in the therapy unit machine has replaced the use of block trays placed over the patient to shape the target treated. Multileaf collimators shape the radiation beam precisely around the tumor target volume and avoid normal tissue. Sophisticated methods by which to move the irradiation delivery device (linear accelerator gantry) around the patient while shaping the field dynamically have also evolved including the use of rotational treatment plans involving the moving of multileaf collimators during rotation. There are multiple successive approaches to contour treatment volume. Another category of physical approaches to optimize the therapeutic ratio includes the use of proton beams, which spare deep tissues by reducing exit dose, as well as, sparing skin and superficial tissues, electron beams to spare deep tissues during superficial target radiotherapy, and high energy particle beam radiotherapy delivery systems (including carbon atoms). The physical modifications in clinical radiotherapy treatment devices have enhanced the precision of dose delivery to clinical target volumes and facilitated radiation dose escalation while maintaining the same acceptably low level of normal tissue damage.

A third category of physical parameters for improvements in radiotherapy involves dose and fraction size [5,6,7]. Hypofractionation (reduced number of treatments including stereotactic radiotherapy and stereotactic body radiotherapy) with higher dose per treatment was tested and found in some cases to produce the same outcome as found with more cumbersome multiweek conventional fractionation schedules, which traditionally deliver 1.8–2.0 Gy per day. Another strategy to spare normal tissues, particularly, in previously irradiated areas, includes hyperfractionation in which multiple small doses of irradiation can be delivered in one day separated by 4–6 h. Thus, multiple types of improvements in the physical parameters of modern radiotherapy continue to optimize the therapeutic ratio.

An entirely independent approach to improving the therapeutic ratio involves biochemical and molecular-biologic strategies. Prominent in this approach has been the development of tumor radiation sensitizers [8], or parenterally delivered drugs that enter the tumor volume and facilitate greater radiation killing of tumor cell numbers with little or no additional normal tissue damage. Selective uptake of a radiosensitizer drug by cancer cells relative to cells in the surrounding normal tissue volume has been the great challenge to improve the therapeutic ratio. A second challenge was the observed toxicity of some radiosensitizer drugs for normal cells and tissues in organs outside the irradiated volume [8].

An alternative biologic strategy for improvement of the therapeutic ratio is the use of radiation protectors for normal tissues [9]. While this strategy can also improve the therapeutic ratio, the reverse challenge is apparent. Delivery of an agent to protect normal tissue might also protect the tumor and obviate any improvement in the therapeutic ratio. With respect to delivery of a radiation protector, this strategy requires design of a safe delivery method. One such approach was the potential use of gene therapy to deliver a radiation protector protein to normal tissues.

### 1.1. The Logic of Utilizing Radioprotective Gene Therapy in Radiation Oncology

Faced with the challenge of treating a large tumor volume, multidisciplinary approaches include surgery, radiation therapy, and systemic chemotherapy. Surgical excision of a large tumor may result in incomplete resection with positive margins and multiple positive regional lymph nodes. The design of radiotherapy treatment volumes may lead to large treatment fields and increase the challenge of achieving a safe therapeutic ratio. Maximum effective radiotherapy doses often have a limiting toxicity usually attributable to one or more organs in the irradiated volume [10].

The data from analysis of organ and tissue specific dose limiting toxicity in radiotherapy presented an opportunity for organ-specific radioprotective gene therapy.

The first studies of organ specific gene therapy for radioprotection were carried out with lung and involved an animal model of thoracic irradiation induced pulmonary fibrosis [11]. Intratracheal injection of a therapeutic gene seemed logical, because airway distribution of the protective agent could be limited to avoid its entry into the peripheral blood circulation and avoid potential protection of tissues outside the irradiation volume [12,13]. The first questions in this approach included (1) finding a safe a reliable vector to deliver the transgene and (2) finding the best transgene to use.

### 1.2. Cell Cultures Experiments Identify the Importance of Mitochondrial Targeting

One of the first discovered irradiation induced cell death pathways was apoptosis [14,15]. Radiation induced nuclear DNA double strand breaks, which were initially thought to be the mechanism of cell death turned out to be the first step in a cascade of events leading to events in the cytoplasm specifically in organelles, (the mitochondria). In the 1970s, alkaline sucrose gradient studies were carried out on ionizing irradiated cells in culture and fragmented DNA was shown to be directly increased by irradiation dose and associated with cell death. The dominant methodology for measuring irradiation induced killing at that time was the clonogenic survival curve [16,17]. Decades of subsequent research demonstrated that these initial nuclear DNA double strand breaks were repaired within minutes, and that the DNA fragmentation associated with death occurred hours to days later, and after signaling between nucleus and mitochondria initiated a cascade of events in the apoptosis cell death pathway [14,15]. Prominent in this biochemical mechanism of apoptosis was the translocation from nucleus to mitochondria of pro-apoptotic signaling molecules including p53, p21, BAX, and others [15]. At the same time that this pro-apoptotic signaling was occurring, cells activated a defense system to prevent cell death. Prominent in the anti-apoptotic response was the upregulation of RNA and protein for manganese superoxide dismutase (MnSOD) [18,19].

MnSOD was one of three general categories of superoxide dismutases identified by biochemists. The metalloenzyme for MnSOD utilized the element manganese in the active site, whereas the other two superoxide dismutases utilized copper [15]. While all forms of SOD including cytoplasmic CuSOD (SOD1) and extracellular SOD (SOD3) had the same biochemical functions of dismutating superoxide to hydrogen peroxide [19], the localization of MnSOD (SOD2) to the mitochondria provided the greatest radiation protection [14,15]. Mitochondrial localization of SOD for effective radioprotection was demonstrated in experiments in which the mitochondrial localization signal (sequence) on the protein of MnSOD was attached to Cu/ZnSOD in transfection experiments using the transgene for this manipulated enzyme. Effective radiation protection was realized with a Cu/Zn metallo-enzyme localized to mitochondria [15]. The reverse experiment in which the mitochondrial localization signal of MnSOD was removed followed by transfection of this altered construct into cells resulted in cytoplasmic MnSOD activity without mitochondrial localization and less radioprotection. Thus, mitochondrial localization provided radiation protection, and that was organelle specific [15].

Other transgene therapies focused on other elements of the initial antioxidant and anti-apoptotic response of cells. Other transgenes included those for glutathione peroxidase, and catalase [20] was tested in transfection experiments with cells in culture and provided some radioprotection, although, not as effective, as was MnSOD [19]. Thus, the transgene for MnSOD with its mitochondrial localization appeared to be an ideal candidate for radioprotective gene therapy.

The next challenge was that of identifying a vector or method by which to insert a transgene into cells, facilitate translocation to the nucleus, and replication of the transgene to produce RNA and protein.

## 2. Plasmid Liposome Vector Transfer of MnSOD for Gene Therapy

Gene therapy techniques have evolved over the past decades and utilized molecular biologic and virologic data that have been accumulated from different sources. Initial studies with retrovirus were based upon decades of research with murine RNA, type-C tumor viruses showing efficient entry into cells, and through reverse transcriptase production of a cDNA leading to production of multiple copies of the RNA. Concern about sites of integration of retroviral cDNA into nuclear DNA led to failed clinical trials, most prominently, the induction of leukemia by retrovirus insertion into cells proximate to oncogenes [1,2,3].

DNA viruses provided an alternative and less dangerous approach [21]; however, inflammatory response to adenovirus vectors also caused fatality [1]. The limitation of size of transgene insert presented a problem. Other studies with Herpes virus provided positive results [22].

The strategy of using plasmid liposomes to carry transgene inserts was an attractive, and safe strategy, since plasmids utilized to deliver the transgene could be precisely controlled with no replication. Cationic liposomes showed sufficient binding to the extracellular membrane and were first explored, as a vehicle by which to deliver plasmids containing MnSOD transgene [12,23]. Initial studies with lung delivery by intratracheal injection [12] or inhalation administration [24] led to effective transgene uptake in the lung. Identification of the cellular phenotypes associated with effective gene therapy was facilitated by attaching an epitope-tag to the MnSOD transgene for localization of its immunogenic protein in pulmonary epithelial, endothelial, and monocyte/macrophage immunocytes in the lung [25,26].

### 2.1. Strategies for Plasmid Liposome Construction

The first use of MnSOD-plasmid liposome (MnSOD-PL) gene therapy was in the model of irradiation induced pulmonary fibrosis [11]. The C57BL/6J mouse model develops a predictable organizing alveolitis/fibrosis initiating at around 100 days after 20 Gy thoracic irradiation [11,12,27]. With this model and using separation techniques to isolate pulmonary epithelial compared to endothelial and infiltrating immunocytes, it was possible to target an epitope-tagged transgene product to different cell types after intratracheal injection of MnSOD-plasmid liposomes [26]. The construction of the plasmid and the choice of liposomes depended initially upon quickness of uptake and ease of intratracheal injection. A plasmid construct was chosen to provide for a robust expression of the MnSOD transgene. Given the size of this cDNA for MnSOD, this construct was initially inserted into the first plasmid chosen for these studies [12]. The detection of plasmid and insertion into multilamellar liposomes followed standard procedures at that time. One challenge was the volume of 100 microliters required for the intratracheal injection and the ease of administration [12]. Techniques by which to carry out intratracheal injection without the need for surgical exposure of the trachea and/or anesthesia of animals was desirable in these first experiments. Plasmid liposome constructs were chosen from experiments with in vitro transfection of cell lines. The hematopoietic progenitor cell line 32D cl 3 [28] was chosen as the test system for comparison of different liposome formulations including lipofectin, lipofectamine, and other liposome formulations commercially available at the time [14,15]. A minicircle plasmid also delivered therapeutic levels of MnSOD [29].

Lipofectin was effective at delivery of microgram quantities of plasmid to the lungs. Tracing plasmid to the lungs was initially carried out by rt-PCR quantitation of plasmid specific sequences in whole lung, and then in sorted fractions of different cell phenotypes within the lung [25,26].

The first in vivo transfection experiments included as controls, plasmid without the MnSOD insert, and empty liposomes alone. These experiments showed that plasmid cDNA was detectable in the lung immediately after injection and persisted for several days. The kinetics of detectable increase in pulmonary MnSOD levels showed a delay between the time of plasmid liposome intratracheal injection and detection of increased levels of MnSOD in tissues. This delay was consistent with that observed in the in vitro experiments with 32 D cl 3 cells [14,15]. The expected delay was that associated with plasmid attachment to the cell membrane, transit through the cytoplasm to the nucleus, penetration of the nucleus, and uncoding of the liposome, cDNA mediated production of large quantities of transgene mRNA, exit of mRNA from the nucleus, and then production of protein and migration of protein to the mitochondrial membrane. In vitro experiments, which established the importance of mitochondrial targeting for effective radiation protection were carried out using a strategy of removing the mitochondrial localization sequence from the MnSOD transgene and attaching it to the cytoplasmic expressed Cu/ZnSOD [15]. Inserting these new constructs into plasmid liposomes and transfecting 32D cl 3 cells in vitro revealed robust radiation protection when mitochondrial targeting was included in the construction of the transgene and coded protein. Mitochondrial localization of Cu/ZnSOD or MnSOD was required for radiation protection [15]. The companion experiment in which the mitochondrial localization peptide sequence was removed from MnSOD resulted in cytoplasmic expression of MnSOD and no radiation protection [15]. These experiments established the time delay required but did not foretell the expected effectiveness of intratracheal MnSOD-plasmid liposome administration for radioprotection.

In other experiments, gene therapy using MnSOD-PL prior to irradiation was clearly effective for radiation protection of the lung, but administration of the construct after irradiation was relatively ineffective [30]. Due to the delay in getting transgene encoded protein to the mitochondria, the administration after irradiation was ineffective since 18 h was required for transgene protein delivery to the mitochondria and this time also associated with significant apoptosis, inflammatory cytokine expression, and stress response gene mediated cellular and tissue damage that follow whole lung irradiation.

### 2.2. Intraesophageal Delivery of MnSOD-Plasmid Liposomes

Initial experiments with whole lung (thoracic) irradiation demonstrated that animals also had significant esophagitis [23]. During the 1990s, radiation dose escalation for the management of patients with non-small cell lung cancer was being carried out at many medical centers, and significant esophagitis was detected in patients receiving thoracic doses exceeding 50 Gy [10,31]. Accordingly, emphasis was shifted to intraesophageal administration of MnSOD-PL constructs. A model of esophagitis demonstrated several reproducible and quantitative parameters of esophageal damage, principally, detection of apoptotic bodies in the esophagus epithelium at 1–3 days after single fraction radiation to the esophagus [23]. The model of irradiation esophagitis was utilized to deliver a swallowed 100 microliter volume of the same plasmid liposomes. Cationic liposomes, which showed robust attachment to pulmonary epithelium also facilitated attachment to esophageal squamous epithelium in the mouse model. Initial experiments were carried out with single fraction [23], but then fractionated irradiation [32], and in conjunction with combination chemotherapy used to treat non-small cell lung cancer. This model of esophageal radioprotection with MnSOD-PL was highly reproducible and the swallowed administration required no anesthesia and easy oral administration to non-anesthetized mice.

The model of esophagus radioprotection was easily translated to clinical radiotherapy patients. In consultation with the FDA and development of an IRB-approved protocol, a second animal species was required, and these studies were carried out in rabbits that had an esophagus similar to humans and with similar development of radiation esophagitis [33]. MnSOD-PL was effective in patients, as well as, in the mouse model and significantly reduced the objective and subjective parameters of esophagitis, most easily quantitated was the number of apoptotic cells per high power fields in standard histopathologic specimens. For both the intrapulmonary injection and the intraesophageal injection of MnSOD-PL, no significant radioprotection was observed with delivery of plasmid liposomes containing no transgene or delivering empty liposomes. For both the pulmonary radiation protection and esophagus radiation protection models, correlates of effective gene therapy included abrogation of the irradiation induced mRNA for inflammatory cytokines and stress response genes, and abrogation of the increase in inflammatory proteins IL-1, TNF-α, and TGF-β [34,35]. Of great interest in both the lung and esophagus radioprotection models was the observation that acute radiation protection was linked to significant reduction of late radiation fibrosis. While fibrosis was the primary endpoint measured in the lung model, esophageal stricture as a secondary endpoint was also quantitated and shown to also be significantly reduced in the acute esophagus radioprotection model [32].

For both the intrapulmonary and intraesophageal injection models, there was concern that MnSOD-PL normal tissue protection would also protect tumors. Orthotopic Lewis Lung carcinomas in the mediastinum of C57BL/6J mice showed no significant uptake of MnSOD-PL when intratracheal injection was carried out or the lung radiation protection model [32]. There was no significant uptake of MnSOD-PL in orthotopic mediastinal tumors in animals given intraesophageal/swallowed MnSOD-PL. These experiments provided important information for the FDA to approve a Phase I Clinical Trial of delivery of MnSOD-PL twice weekly in patients with unresectable Stage IIIA/B non-small cell carcinoma or esophageal radioprotection [33]. While the MnSOD transgene was easily detectable in mice given 100 microliters of swallowed MnSOD-PL, biopsy of the esophagus at several locations and at several time points during the clinical protocol revealed no detectable MnSOD transgene. The results were not surprising given the volume of swallowed MnSOD-PL in patients receiving five fractions of radiotherapy per week, and a twice weekly swallow of MnSOD-PL. The reason for the twice weekly administration was based on studies in the mouse esophagus in which the MnSOD transgene encoded RNA and gene product was detectable for 48 h. Similar findings in a rabbit model indicated that twice weekly administration would be sufficient to keep MnSOD transgene levels elevated, and keep transgene encoded protein constantly at the mitochondria in cells of the human esophagus to provide radiation protection.

Esophageal radioprotection for lung cancer patients was planned for a large Phase II Clinical Trial; however, data from [9], as well as a single institution study [31] cooperative RTOG trial indicated that 70 Gy was no longer required for effective local control of unresectable non-small cell lung cancer, given the effectiveness of new chemotherapy drugs including Carboplatinum and Taxol. In addition, the RTOG-0617 study [9] indicated that there was significant comorbidity with radiation doses above 60 Gy. Improved treatment planning techniques including Intensity Modulated Radiotherapy (IMRT) and respiratory gating have minimized the length of esophagus in the irradiation field. Therefore, the radioprotective gene therapy strategy for the esophagus was felt to be unnecessary.

### 2.3. MnSOD-Plasmid Liposome Gene Therapy of the Oral Cavity and Oropharynx

Intraoral administration of MnSOD-PL was shown to be highly effective in preventing oral cavity mucositis and salivary gland toxicity in the mouse model of head and neck irradiation [36,37,38,39,40]. Mucositis was a dose limiting and prohibitive toxicity in patients receiving chemoradiotherapy for squamous cell carcinoma of the head and neck region. Patients with advanced stage disease, and those with multiple positive lymph nodes required large volume head and neck irradiation and were ideal candidates for radiation protection strategy for normal tissues.

The C57BL/6J mouse model of oral cavity radiation mucositis was chosen to test the effectiveness of intraoral administration of MnSOD-PL for single fraction and fractionated irradiation of the head and neck region [36,37,38,39,40]. In multiple mouse experiments, including those with orthotopic floor of the mouth cancer, intraoral MnSOD-PL was shown to be protective of normal oral cavity tissues, while not preventing radiation mediated local control of tumor [37]. Interesting in these studies was the effect of intravenous administration of MnSOD-PL, which demonstrated effective uptake of epitope-tagged transgene product in the tumor, as well as, in the oral cavity. This result was expected and was opposite to the absence of detectable transgene product in tumors when oral administration was delivered. Floor of the mouth orthotopic cancers had no contact with the oral mucosa, and, thus, penetration of transgene through multiple layers of oral mucosa into the tumor was not detected [37,38,39,40].

Of interest, the MnSOD-PL transgene was detected in the tumors after I.V. administration to animals with orthotopic tumors and showed a tumoricidal effect [37]. This result was consistent with the publications from Larry Oberley and colleagues [18] indicating that squamous cell tumors, which are often deficient in glutathione peroxidase, cannot effectively tolerate the hydrogen peroxide generated by MnSOD mediated dismutation of superoxide to hydrogen peroxide. The data were consistent with translation of intraoral administration of MnSOD-PL to the clinic. Due to the rapidly emerging availability of MnSOD mimetic molecules with more rapid delivery of mitochondrial protecting molecules directly to mitochondria, the MnSOD-PL strategy was replaced by a program using GS-nitroxide small molecules in the clinic for oral cavity radioprotection.

### 2.4. Systemic Intravenous Administration of MnSOD-Plasmid Liposomes

The delivery of MnSOD-PL intravenously was carried out as part of the control experiments for the intraoral administration in mice with orthotopic tumors of the floor of the mouth [37]. In these experiments, it was determined that intravenous administration of MnSOD-PL was safe and effective in delivering the drug to multiple tissues. Systemic administration of liposomes was effective in the clinic in other settings including the delivery of chemotherapeutic drugs with targeted intravascular delivery to specific organ sites. Experiments demonstrating the effectiveness of MnSOD-PL as a radiation protector (when delivered before irradiation) or prior to each of multiple fractions of radiotherapy, were not effective delivering MnSOD-PL after irradiation. Furthermore, for use in clinical radiotherapy for cancer patients I.V. delivered radiation protectors (despite the hypothesized generation of hydrogen peroxide) and ineffectiveness of tumor cells to manage hydrogen peroxide was not felt to be consistent with the strategy of improving the therapeutic ratio. Therefore, intravenous administration of MnSOD-PL was not developed.

### 2.5. Time Course of Administration of MnSOD-Plasmid Liposomes as Radioprotective Gene Therapy

In vitro experiments demonstrated that high level administration of MnSOD-PL to cells in culture resulted in detection of elevated levels of MnSOD protein at around 18 h after administration. Given the high concentration of plasmid liposomes deliverable to cells in culture and the relative homogeneity of delivery to lung, esophagus, and oral cavity, it was not surprising that experiments to deliver MnSOD-PL after irradiation exposure were relatively ineffective. Swelling of irradiated tissues, compromised vasculature, and the rapid increase in levels of inflammatory cytokines and stress response genes within normal tissues after irradiation made the local delivery or intravenous delivery of MnSOD-PL extremely unlikely to result in high level transgene penetration into cells, and rapid production of protein. The time course was felt to be relatively prohibitive and a negative for the use of MnSOD-PL in radiomitigation.

Due to the time delay in getting MnSOD protein to the mitochondria, we concluded that there was a relative ineffectiveness of MnSOD-PL gene therapy for systemic radiation mitigation [30].

### 2.6. A Compensatory Mechanism for Response of Tissues to MnSOD-PL Gene Therapy

The response of tissues and organs to irradiation is one potential method of activation of several inflammatory responses and cell death pathways. Rapid elevation in levels of MnSOD was clearly radioprotective, but multiple administrations were shown to be no more effective than the single administration prior to irradiation.

One of the conclusions from these experiments was that there might be an adaptation response of tissues in cells to the introduction of high levels of one radioprotective protein and potentially the downregulation of others. To confirm these hypotheses, experiments were carried out with transgenic mice showing a constitutive increased expression of MnSOD. Other experiments were carried out with MnSOD homologous recombination negative (knock-out) mice [26].

Transgenic mice overexpressing MnSOD were not intrinsically radioresistant [41].

These experiments suggested that transgenic animals had adapted during gestation. Experiments with homologous recombinant negative MnSOD (knock-out) mice demonstrated significant radiosensitivity of the animals and cell lines derived from the bone marrow microenvironment of such mice. In contrast to the studies with overexpression of MnSOD in transgenic mice, the knock-out experiments demonstrated the critical importance of MnSOD during development of genetically deficient animals, particularly during the gestational period.

Radiosensitivity of MnSOD knock-out mice was reversed by insertion of the transgene for MnSOD into cells in culture and also administration of MnSOD-plasmid liposomes restoring normal tissue balance [26]. These experiments proved important for documenting to the FDA the relative safety of the administration of MnSOD-plasmid liposomes since genetically deficient animals showed a restored antioxidant balance and radiosensitivity by reintroduction of the transgene to produce normalized levels of MnSOD.

Experiments with both genetic homologous recombinant negative, and with overexpressing transgenic mice led to the conclusion that adaptation to overexpression or deficiency of a specific antioxidant protein may in some conditions lead to compensatory downregulation or upregulation of other antioxidant mechanisms.

### 2.7. Challenges Regarding the Effectiveness of MnSOD-Plasmid Liposome Gene Therapy Compared to Other Radiation Protection Strategies

The plan for gene therapy for radiation protection initially seemed to be well supported by biochemical mechanisms of continuous production of the transgene product for several days in cells in culture and also in animal models. The strategy of a single administration of a transgene rather than need for continuous administration of multiple injections or continuous oral intake of an antioxidant radioprotector seemed logical. However, there were complications regarding gene therapy, some of which persist to the present day (Figure 1).

Even though the vector for transfer of the transgene was a plasmid liposome construct with no significant risk of prior application or transgene insertion into the somatic cell genome, there were concerns about gene therapy in general. There is a greater acceptance of the possibility of using gene therapy to deliver a toxin to a cancer, rather than a protector to normal tissues. The logic was that the cancer cells were already abnormal and needed to be eliminated, so an unexpected toxic effect in the cancer cells would be acceptable. In contrast, an unanticipated toxicity in normal tissue would not be acceptable. During the process of getting FDA approval for a Phase I Clinical Trial of esophagus radiation protection in unresectable non-small cell lung cancer patients, FDA examiners requested a list of the potential complications of the administration of MnSOD-PL to normal tissues. Since there were no detectable deleterious effects of empty liposomes or administration of plasmid, and since there were no deleterious effects of overexpression of MnSOD in transgenic animals or in the experimental gene therapy model systems, it was difficult to determine what a possible side effect of gene expression in normal tissues might be. In a clinical protocol, the listed potentially unexpected toxicities included neutropenia and cytopenia, and other hematological abnormalities. Since the first gene therapy trial using MnSOD-PL was to administer the drug by plasmid liposomes swallowed and going through the esophagus into the stomach and since the transgene and liposomes would be expected to have total degradation in the stomach, there were no anticipated side effects regarding esophagus, stomach, or intestine [33]. One possible unexpected toxicity was thought to be the expression of the MnSOD transgene in the esophagus leading to overexpression of other compensatory genes or downregulation of other antioxidant gene products.

Alternatives to MnSOD-PL gene therapy for radiation protection are abundant. Delivery of MnSOD or other SOD proteins directly into tissues was carried out with some success. The administration of other agents to minimize radiation toxicity and provide radiation protection includes the antioxidant defense system in cells that is composed of not only superoxide dismutase, catalase, glutathione peroxidase, and other antioxidant enzymes, but also small molecule radiation protectors. Cellular glutathione is a major biochemical antioxidant and radioprotector, which is depleted rapidly after radiation exposure. Administration of agents designed to increase glutathione concentration in cells was an alternative strategy for radioprotection [20]. Other methods by which to neutralize irradiation induced free radicals or reactive oxygen species (ROS) included administration of sulfhydryl compounds and small molecule antioxidants [18,42,43]. Prominent as a small molecule radiation protector, Amifostine was delivered in clinical trials to provide normal tissue radioprotection in several settings [44]. Success was achieved in intraoral administration of Amifostine for patients receiving radiotherapy for head and neck cancer. Concentration of the compound in the salivary glands was detected, and, thus, the amelioration of xerostomia associated with clinical radiotherapy was a prominent result of this clinical trial. However, as with many other small molecule radioprotectants there were side effects of Amifostine, and these included hypotension and renal toxicity. The ease of administration of a small molecule, however, was a potential clear advantage to the use of gene therapy and certainly less expensive. The expense in producing transgene, packaging in liposome vehicle/vector, and assuring uniformity between batches/lots of drug was a challenge for the development of MnSOD-PL gene therapy.

## 3. Development of MnSOD Mimetics

Mitochondrial targeting for the effectiveness of MnSOD led to the conclusion that mitochondrial specific delivery of antioxidant drugs might be an alternative to MnSOD-PL gene therapy. Nitroxides were demonstrated to be outstanding as potential antioxidants [42]. The nitroxide cycles through a hydroxyl amine phase and continually neutralizes free radicals. Mitochondrial targeting of nitroxide to enhance its effectiveness as a radiation protector was also found to be potent for radiation mitigation, and the timing of delivery could be gauged to be more efficient than the 18 h required for MnSOD transgene product to be synthesized and transported to the mitochondria. Mitochondrial targeting of nitroxide was achieved through use of the hemigramicidin peptide isostere and also using a Triphenyl Phosphonium structure both of which led to significant increases in nitroxide within the mitochondrial membrane [42,45]. Variation in the degree of mitochondrial specific concentration was found to vary between the XJB-5-131 moiety, which produced a 400–600-fold increase at the mitochondrial membrane compared to the smaller JP4-039, which produced a 33-fold increase in mitochondrial targeting [42]. Both of these agents proved to be superior radiation protectors and radiation mitigators. Systemic delivery of these water-soluble compounds posed a challenge very similar to that for systemic delivery of MnSOD-PL. Liposomal packaging of the GS-nitroxides was achieved using multilamellar liposomes (F14 and F15) [39,40,46]. These techniques allowed delivery of the small molecule to esophagus, lung, and oral cavity providing similar radiation protection and mitigation to that of previous studies with MnSOD-PL.

## 4. Conclusions and Future Challenges

Radiation protection and mitigation requires continuous attention to detail regarding the therapeutic ratio. In the scenario of the cancer patient who may require high dose radiotherapy to a relatively large target volume, normal tissue protection must be achieved, but with no significant protection of the tumor. In contrast, with a scenario of radiation counter-terrorism or the mitigation of normal tissue damage following a radiation accident, distribution of a small molecule MnSOD mimetic needs to be uniform and based on mitochondrial targeting and requires design of a drug that can be delivered safely with minimal toxicity. Despite the change in direction from a MnSOD gene therapy approach to one of a MnSOD mimetic, these challenges will persist. Studies with MnSOD transgene therapy did lead to discoveries that facilitated development of small molecule MnSOD mimics and all of this work was based upon the overarching discovery of the importance of mitochondrial targeting.

The advantages of using MnSOD-PL as a radioprotector is that it has been safe to use not only in our animal models but also in clinical trials. The MnSOD plasmid is not incorporated into the genome but is lost from the cell in 48–72 h so increased expression of MnSOD only occurs for a short period of time and then returns to normal levels of expression. One of the problems with using MnSOD-PL is that it must be given 24 h before irradiation to allow time for optimal expression of MnSOD before irradiation. The use of MnSOD-PL for radiation mitigation has not been effective since it takes 24 h to be effective. Future research would be to use a new gene promoter to allow for faster expression of MnSOD so that it does not have to be given 24 h before irradiation which may allow for it to be used as a radiomitigator. Additionally, the use of a different liposome may allow for more efficient uptake of the MnSOD plasmid into more cells to allow for greater radioprotection.

## Figures and Tables

**Figure 1 antioxidants-10-01057-f001:**
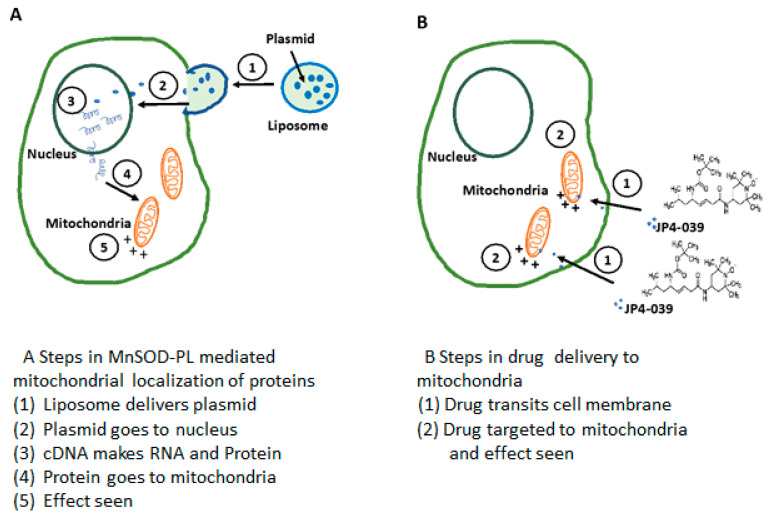
Transgene expression of plasmid/liposome complex or small molecules for radioprotection or radiomitigation. (**A**) shows the steps involved in expression of MnSOD from MnSOD-PL for radioprotection. (**B**) demonstrates the steps in the delivery of small molecule radiomitigator JP4-039 to the mitochondria.

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
