# Peer review of "Gene Therapy for Systemic or Organ Specific Delivery of Manganese Superoxide Dismutase"

_antioxidants, 2021, doi:10.3390/antiox10071057_

Round 1

Reviewer 1 Report

The manuscript entitled “Gene Therapy for Systemic or Organ Specific Delivery of Manganese Superoxide Dismutase”, authored by Joel S. Greenberger, Amitava Mukherjee and Michael W. Epperly, deals with the reviewing the potential use of different delivery systems of MnSOD as potential gene therapy. The topic is very interesting, and in general the review is well organizeted. However, small changes, mainly related to the formatting of the manuscript than to the contents, should be fixed.

In particular:

  • authors should add more than three keywords. Furthermore, keywords should be words not contained in the title of the manuscript, and limitedly present in the abstract. I strongly suggest authors to introduce more keywords (maximum 10) and replace those that are repeated in the title. The usefulness of keywords is to make the article both more and more easily searchable visible after its publication through commonly used search engines.
  • It is not clear how the references were included in the manuscripts. They should follow an ascending order according to their appearance in the text.
  • Authors should consider to include the purpose of the review at the end of the abstract section. Indeed, by the simple reading of the abstract, it is not clear what the purpose of the manuscript is. This is a regret, because the revised manuscript has a great potential, which is greatly underestimated by the abstract. Considering that the abstract can contain a maximum number of 200 words, the authors will have to minimally modify the abstract in order to include this essential part.
  • Authors should strongly consider organizing the review into sections that differ from the introductory one. For example, a possible structure could be the following: 1. Introduction; 1.2. The logic of utilizing radioprotective gene therapy in Radiation Oncology; 1.3. Cell culture experiments identify the importance of mitochondrial targeting; 2. Plasmid liposome vector transfer of MnSOD for gene therapy. 2.1. Strategies for plasmid liposome construction; 2.2. Intraesophageal delivery of MnSOD-plasmid liposomes; 2.3. MnSOD-plasmid liposome gene therapy of the oral cavity and oropharynx; 2.4. Systemic intravenous administration of MnSOD-plasmid liposomes; 2.5. Time course of administration of MnSOD-Plasmid Liposomes as radioprotective gene therapy; 2.6. Compensatory mechanism for response of tissues to MnSOD-PL gene therapy; 2.7. Challenges regarding the effectiveness of MnSOD-plasmid liposome gene therapy compared to other radiation protection strategies; 3. Development of MnSOD mimetics; 4. Conclusion and Future challenges.
  • The panels constituting figure 1 should be replaced horizontally. The caption of the figure should precede the figure itself. Furthermore, the caption should have a title, and subsequently the dicitirua describing what is shown in the individual panels.

Author Response

Reviewer One

We want to thank the reviewer for his insightful comments and revised our manuscript as indicated below.  We also want to thank the reviewer for his enthusiasm for this manuscript.

  1. Addition of new keywords:  We have added new keywords such as radioprotection, radiomitigation, plasmid/liposome complex, mitochondrial targeting and MnSOD mimetics.  We have removed keywords found in the title.
  2. References need organized and should be in ascending order: We have renumbered the references beginning with 1.  We also have removed the references which were not used in the text.
  3. Add the purpose of the review to the abstract: We have modified and shortened the abstract so that we can add a statement on the purpose of the review.
  4. Reorganization of the review: We have reorganized the review as suggested.
  5. Place the panels of Figure 1 horizontally and add title:  A title to the figure has been added and a description of the figure has been added.  Figure 1A and 1B are now positioned horizontally.

Reviewer 2 Report

The reviewed manuscript entitled "Gene Therapy for Systemic or Organ Specific Delivery of Manganese Superoxide Dismutase" is out of the scope of the Journal's aims. First, it looks like a part of the thesis - see how the references were presented. Second, there is a very week connection between antioxidant behavior and described here topic. In my opinion, the manuscript is more relevant to journals connected with oncology. Nevertheless, even the manuscript will be transferred to another journal a lot of improvements are necessary.
First, state is an overview or presented results in the abstract.
l. 11 -During the course of development and clinical application of this program - change the sentence or explain what kind of program?
Clearly explain that the manuscript is a review.
l. 19 - "Lessons learned during..." this sentence is difficult to understand - rewrite it or remove it - here there are no classes but should be a comprehensive review,
l. 24 - 38 more references should be added
Please change the reference numbers starting from 1, ... as is recommended in the template.
For a better presentation use graphs, tables and figures. Here only 2 were presented.
Only 10 pages from 23 were dedicated to presenting the aim of the paper, the rest were references that were difficult to find in the manuscript.
Overall, the manuscript should not be presented in the Antioxidants.

Author Response

Reviewer Two

We want to thank the reviewer for his comments.  We have revised the manuscript as suggested and believe it is now a better manuscript.

  1. State in an overview in the abstract:  We have placed a purpose to the abstract which clearly shows that the manuscript is a review.
  2. 1.11 change the sentence or explain what kind of program: The sentence has been revised to clearly identify what the program is.
  3. L. 24-38 more references:  We had added more references describing therapeutic ratio.
  4. Change reference numbers: We have renumbered the references.
  5. References are hard to locate: We have reorganized the references and removed many of the references.  There are now only 46 references.

Reviewer 3 Report

This overview describes the provision of optimal radiation protection during radiation therapy. It is well written on the basis of 197 properly selected literature references.

However, authors in conclusions should write:

1)    what are the advantages and disadvantages of the radiation protection methods used;

2)    what further research should involve to further explore the issue under consideration

Author Response

Reviewer Three

We want to thank the reviewer for his comments on our manuscript.  We have revised the manuscript according to the reviewer’s suggestions and believe that the manuscript is greatly improved.

  1. In the conclusions describe what are the advantages and disadvantages of the radioprotection methods described:  We have added advantage and disadvantages in the last paragraph of the conclusions.
  2. What further research should be explored:  Areas of research which may aid the use of MnSOD-PL have been listed.

Round 2

Reviewer 2 Report

I carefully read the revised version of the manuscript. Still, in my opinion, the presented idea is weak to defense in the scope of the Journal. Still, I can find a clear connection between the presented topic and antioxidant behavior. For that reason, I m recommending shifting this manuscript to some other more focused journals. Some additional comments are listed below:

Abstract

  • "several prominent observations were made including" - by who? (the Authors or some references)

Figure 1 - enlarge the chemical formula, the figure caption should be below not above the figure

What is more, only 46 references for a review is like minireview and discussion, in my opinion, should be extended for more references. Some tables which will summarize the most important data and references will strongly increase the manuscript quality.

Overall I m recommending this manuscript to reject.

Author Response

We want to thank Reviewer 2 for his comments and have responded to these comments.  We have added a new paragraph at the beginning of the introduction stressing the importance of the antioxidant role of MnSOD in protecting normal tissue from irradiation damage.  Hopefully, this shows why we think this manuscript is appropriate for Antioxidants.  In the abstract we have named a some names to identify the who.  We have also increased the size of the chemical formulations in figure 1.  We now hope this manuscript is appropriate for publication in Antioxidants.